# Five-Year Monitoring of a Desert Burrow-Dwelling Spider Following an Environmental Disaster Indicates Long-Term Impacts

**DOI:** 10.3390/insects13010101

**Published:** 2022-01-17

**Authors:** Efrat Gavish-Regev, Igor Armiach Steinpress, Ibrahim N. A. Salman, Nitzan Segev, Assaf Uzan, Yebin Byun, Tanya Levy, Shlomi Aharon, Yoram Zvik, Raisa Shtuhin, Yotam Shapira, Marija Majer, Zeana Ganem, Sergei Zonstein, Ivan L. F. Magalhaes, Yael Lubin

**Affiliations:** 1The National Natural History Collections, Edmond J. Safra Campus, Givat Ram, The Hebrew University of Jerusalem, Jerusalem 9190401, Israel; igor.armiach@mail.huji.ac.il (I.A.S.); assafuzan@gmail.com (A.U.); yebin.byun@mail.huji.ac.il (Y.B.); levy.tanya7@gmail.com (T.L.); shlomi.aharon1@mail.huji.ac.il (S.A.); Yoram.Zvik@mail.huji.ac.il (Y.Z.); raisa.shtuhin@mail.huji.ac.il (R.S.); yotam.shapira@mail.huji.ac.il (Y.S.); zeana.ganem@mail.huji.ac.il (Z.G.); 2The Department of Ecology, Evolution and Behavior, Edmond J. Safra Campus, Givat Ram, The Hebrew University of Jerusalem, Jerusalem 9190401, Israel; 3French Associates Institute for Agriculture and Biotechnology of Drylands, The Jacob Blaustein Institutes for Desert Research, Ben-Gurion University of the Negev, Sede Boqer Campus, Midreshet Ben-Gurion 8499000, Israel; ibrahim.salman92@gmail.com; 4Mitrani Department of Desert Ecology, Ben-Gurion University of the Negev, Midreshet Ben-Gurion 8499000, Israel; mmajer98@gmail.com (M.M.); lubin@bgu.ac.il (Y.L.); 5Dead-Sea & Arava Science Center, Yotvata 8882000, Israel; nitzan@adssc.org; 6The Scorpion Research Lab, Hoopoe Ornithology & Ecology, Yeroham 8051875, Israel; 7Steinhardt Museum of Natural History, Tel Aviv University, 12 Klausner St., Tel Aviv 6139001, Israel; serzon56@gmail.com; 8División Aracnología, Museo Argentino de Ciencias Naturales “Bernardino Rivadavia”—CONICET. Av. Ángel Gallardo 470, Buenos Aires C1405DJR, Argentina; magalhaes@macn.gov.ar

**Keywords:** Araneae, ‘Avrona, ‘Arava valley, bioindicator, hyper-arid, Filistatidae, oil-spill, *Sahastata*

## Abstract

**Simple Summary:**

Deserts are characterized by unpredictable precipitation, extreme temperatures, and plants and animals that are specialized to live in these habitats. Consequently, desert organisms often recover slowly, if at all, from human-induced environmental disasters. We studied the effects of two nearby oil spills from a broken pipeline, one that occurred in 1975 and another recent one in 2014, on a burrow-dwelling spider in the extreme desert of the ‘Arava valley (Israel). We compared the abundance of spider burrows in plots contaminated by the oil with nearby unaffected plots over a 4-year period. The abundance was significantly lower in plots with oil-contaminated soil, both in the recent (2014) oil spill area and in the area affected by the 1975 spill. In the laboratory, we found that when offered oil-contaminated versus clean desert soil substrates, spiders chose the clean soil substrate. We conclude that the populations of this burrow-dwelling spider were affected negatively by the oil spills and, furthermore, showed long-lasting impacts from a 40-year-old spill. We propose that burrow-dwelling spiders can be used as effective bioindicators of persistent soil pollution in desert habitats.

**Abstract:**

Deserts are characterized by unpredictable precipitation and extreme temperatures. Their fauna and flora are sensitive to anthropogenic environmental changes, and often recover slowly from environmental disasters. The effects of oil spills on the biota of desert regions, however, have scarcely been studied. We predicted that terrestrial invertebrates suffer long-term negative effects from an oil spill, due to their close association with the substrate. Thus, we investigated the effects of two oil spills that occurred in 1975 and 2014 in the hyper-arid ‘Arava desert (Israel), on a spider that constructs silk-lined nests in burrows in compact, sandy soil in this extreme desert habitat. The spider, *Sahastata aravaensis* sp. nov. (Filistatidae), is described herein. We assessed spider burrow abundance in plots located in oil-contaminated and nearby uncontaminated clean soil (control) areas over five consecutive years and measured habitat characteristics in these plots. In the laboratory, we determined the preference of individuals for clean vs. oil-contaminated soil as a resting substrate. Finally, as this species was previously undescribed, we added a new species description. The abundance of *Sahastata* was significantly lower in oil-contaminated plots, and this was the case in the 40-year-old oil spill (1975) as well as in the recent one (2014). In laboratory tests, spiders showed a significant preference for the clean soil substrate over the oil-contaminated substrate. In the field, soil crust hardness and vegetation density did not differ significantly between oil-contaminated and control plots, but these measures were highly variable. The burrows were significantly clustered, suggesting that the young disperse only short distances. In the laboratory adult spiders did not dig burrows, perhaps indicating that adults remain permanently in their natal burrows and that in the field they may use vacant burrows. We conclude that *Sahastata* populations were affected negatively by the oil spills and these effects were long-lasting. We propose that by monitoring their spatial distribution, burrow-dwelling spiders such as *Sahastata* can be used as effective bioindicators of soil pollution in desert habitats.

## 1. Introduction

Deserts around the world sometimes differ in their abiotic and biotic characteristics, yet they are all defined primarily by aridity and temperature [1]. Uncertainty of precipitation events and their spatial-temporal inconsistency, as well as their variability in magnitude, makes deserts unpredictable and sensitive habitats with relatively low productivity [1,2]. Due to their extreme natural conditions, deserts and their fauna and flora are sensitive to anthropogenic environmental changes and recover slowly from environmental modifications and disasters [3]. For example, it took ca. 50 years for perennial vegetation in the Sonoran desert of Arizona, to recover from a long period of grazing [4]. Another study reported the very slow re-colonization of scorpions, an important group of desert-dwelling predatory arachnids, after the direct removal of 4500 individuals by a researcher from an area of 4000 square meters. In this study, only a few scorpions were observed in the removal area over 30 years of monitoring following the removal [5,6]. These examples demonstrate that in arid or other harsh environments, extreme or long-term disturbances that directly affect specific taxa or life forms in the ecosystem will be very slow to recover. Large-scale natural or anthropogenic ecological disasters, such as fires, floods, or oil spills, may cause high mortality to fauna and flora over substantial areas. Therefore, it may take long, if ever, to return to the original community after large-scale ecological disasters [7].

Terrestrial oil spills are less well studied than marine oil spills. Where they occur in deserts, oil spills are a major threat to the food-webs, due to their direct long-term effects on the interface between soil, water, and plants as well as on microbial populations [8,9]. Two oil spills occurred within a period of 40 years in the nature reserve of ‘Avrona (also spelled Evrona), in the hyper-arid region of the southern ‘Arava valley in Israel. In 1975, ca. 9000 cubic meters of crude oil flooded the dry stream beds of ‘Avrona due to a broken oil pipeline. The oil layer solidified and was covered with time by eroded sediments [10,11,12]. In December 2014, a second spill took place, with ca. 5000 cubic meters of crude oil flood from the same pipeline, 4.5 km north of the 1975 spill event [12,13]. It was estimated that the oil penetrated between a few millimeters to a maximum of 30 cm in both the 1975 and 2014 sites [10]. Figure 1 shows the contaminated areas of the oil spills of 1975 and 2014. Aside from some oil removal by pumping, no treatment or monitoring was conducted after the 1975 oil spill. After the 2014 oil spill, however, several actions were taken, including oil removal by pumping, tillage, and bio-stimulation of the oil-contaminated soil. In addition, a monitoring scheme was launched after the 2014 oil spill, on both 1975 and 2014 spill areas [14,15].

In 2016, two years after the 2014 oil spill, the Israel Nature and Parks Authority (INPA) and Israel’s National Ecosystem Assessment Program (HaMA’ARAG), began a five-year monitoring project to monitor its effect on the soil, soil bacteria, vegetation, invertebrates (insects and arachnids) and vertebrates (reptiles, birds, gazelles, and bats). The monitoring suggested that persistent, oil-induced soil hydrophobicity over time in both oil spills and the resulting oil crusts, caused high resistance to water penetration [10], which, in turn, negatively affected the vegetation. Acacia trees (several species of the genus *Vachellia* (Wight & Arn.)), the keystone species in this ecosystem, were found to be affected by oil spills [16,17]; germination, seedling height, leaf number, stem diameter, and root growth were significantly reduced in oil-contaminated soils than in clean soils [17]. Moreover, five different metabolites (alanine, β-alanine, acid, maltose, and phosphoric acid) had significantly different levels in the oil-contaminated and clean soils of Acacia trees [16]. Surprisingly, no effect of the oil spill was detected on seed-eating bruchid beetles [18], and only a moderate effect on parasitoid wasps [19].

Ground-dwelling arachnids constitute a main group of small predators in desert ecosystems worldwide. For instance, 31 arachnid species were reported from deserts, with relatively high abundances (3200 ± 8800 per hectare, see Table 7.3 in Polis and Yamashita [20]), and a single spider burrowing species from the hyper-arid Namib desert had 9-302 individuals per hectare (see Table 7 in Henschel [21]). In Israel, 136 spider species were reported from the arid Negev desert [22] and 45 species from the hyper-arid ‘Arava valley [23]. The composition of arachnid species assemblages is affected by both large-scale habitat features (e.g., alluvial fans, hill slopes, loess plains, sand dunes, and wadis) [22,23], as well as microhabitat features (e.g., crust, annual vegetation, perennial vegetation) [19,21]. Oil spills are likely to have strong negative effects on assemblages of ground-dwelling arachnids, yet these have not been investigated to date. Many desert arachnids avoid the harsh climatic conditions by being active mainly at night and hiding under stones, bushes, or in burrows during daytime [24,25]. However, these behaviors and shelters are unlikely to protect them from direct damage to the substrate caused by an oil spill. Oil spills are likely to pose a higher risk to populations of ground-dwelling species than to vegetation-dwelling species, due to the direct effect on their microhabitat.

In this study, we investigated the effect of the two oil spills in the ‘Avrona nature reserve on desert ground-dwelling scorpions and spiders. Here we report specifically on the effects on a burrow-dwelling spider of the genus *Sahastata* Benoit, 1968 [26], a new genus record for Israel, and a species new to science. *Sahastata* belongs to the family Filistatidae, known as the crevice weaver spiders, an araneomorph family of cribellate spiders [27]. The genus *Sahastata* is distributed in arid and semi-arid areas, from westernmost Sahara to India, and includes nine known species [27,28], however, prior to our study, the only record of *Sahastata* from the eastern Mediterranean was from Egypt [28]. As *Sahastata* lives in burrows in the ground, we hypothesized that the severe changes to its habitat, due to the oil spills, will cause a long-term reduction in its abundance in oil-contaminated soil compared to its abundance in areas with clean soil. Furthermore, we hypothesized that the reduction in *Sahastata* abundance is due to indirect effects of the oil on the soil such as greater soil hardness, and to direct effects of mortality due to toxicity and a repellent effect due to volatile compounds of the oil contamination.

Environmental disasters such as oil spills are unplanned, therefore there is often insufficient information about the state of the ecosystem before the disaster [29]. To test the effects of the disaster on the contaminated area, i.e., the treatment, it is impossible to design a balanced field experiment with sufficient replications and with randomized assignments of treatment and control areas. Furthermore, the treatment plots are compared to uncontaminated reference areas, which are not true controls [29]. In this study, we monitored *Sahastata* burrows in the field over five years in oil-contaminated and clean soil reference areas and tested the spider’s response to oil-contaminated soil in laboratory experiments. By using a design that includes repeated monitoring over five years, we could test both long-lasting impacts [29], and by adding laboratory experiments, we could explore specific mechanisms.

## 2. Materials and Methods

### 2.1. Monitoring

‘Avrona nature reserve (29.676556, 35.004639) is characterized by a hyper-arid climate with low mean annual precipitation (30 mm), 2600 mm/year potential evapotranspiration, and mean annual radiation of 20.0 MJ/m^2^/day. The mean annual temperature is 25 °C with a mean daily range of 14 °C, and soil surface average annual temperature that can reach up to 50 °C at noon and drop to 30 °C at night. The mean annual humidity ranges between 40% to 45% [30,31]. In May 2016, we launched a five-year arachnid monitoring project to test the effects of the 1975 and 2014 oil spills on the arachnid assemblages of the nature reserve. In the 2014 spill area, the soil is mainly sand compared to the area of 1975, where the soil is covered by gravel (also called reg or hamada) (Figure 2). The oil flowed in both areas primarily in the main wide wadis, where the keystone species of Acacia trees (*Vachellia* species) are concentrated (Figure 1). During the first year of monitoring (2016), we encountered five spider burrows while installing pitfall traps (Figure 1, white dots). The spider burrows were found in both 1975 (two burrows) and 2014 (three burrows) areas, four of them in the clean soil (control) areas and one in the 1975 oil-contaminated soil, but not on direct oil. All burrows contained cribellate-silk nests (Figure 3). Between 2017–2020 we conducted monitoring of *Sahastata* burrows by means of visual search during daytime within and between 16,100 m^2^ monitoring plots, four of each type: 2014 control, 2014 oil-contaminated, 1975 control, 1975 oil-contaminated. The monitoring plots were chosen to include a similar number of the Acacia tree keystone species. Control plots were situated tens to hundreds of meters in parallel to the oil-contaminated plots so that they would have similar characteristics but were not within the path of the oil spill (Figure 1). The plots were surveyed visually at least once a year (see Appendix B for exact monitoring dates), each by one experienced arachnologist together with one to four additional researchers, by walking back and forth along the plot until covering it completely. In the first-year monitoring, each *Sahastata* burrow encountered in each plot was marked in the application Fulcrum © [32], and the length, diameter, and height of the burrow’s opening were measured and documented using a Vernier caliper or metal ruler. Additionally, the presence of a spider in the burrow was documented. Presence was indicated if the spider responded to stimulating the silk in the burrow entrance with a twig or was inferred if there was fresh cribellate silk around the burrow opening (Figure 4, Appendix A). In the subsequent years, in addition to the marking of new burrows in all plots, we visited previously marked burrows and again documented whether the spider was present in the burrow. A subset of the burrows was re-measured.

### 2.2. Quadrat Survey, Nearest Neighbor, and Habitat Characteristics

To understand the habitat characteristics of *Sahastata*, we compared 38 quadrats of 45 cm^2^ in November 2018. Eighteen of the quadrats included a *Sahastata* burrow, and twenty were adjacent, no-burrow quadrats that were chosen randomly nearby, by throwing a flag with a metal stick. Quadrats were located in both 1975 and 2014 clean soil (control) and oil-contaminated areas. 1975 control: with burrow (6), no-burrow (5); 1975 contaminated: with burrow (4), no burrow (5); 2014 control: with burrow (6), no-burrow (6); 2014 contaminated: with burrow (2), no-burrow (4). Quadrats were documented using a Nikon D7100 camera with Sigma 50 mm Macro lens for further analysis of the percentage of stones and X-Rite Color Checker Passport 2 for color calibration (Figure 5). For each quadrat, we documented soil type and hardness at five points inside each quadrat using a pocket penetrometer and measured the distance of the *Sahastata* burrow, or the center of the no-burrow quadrat, to the nearest *Sahastata* burrow, shrub, tree, and ant nest.

### 2.3. Soil Preference: Laboratory Experiments

On 28 November 2018, we collected eight female spiders from ‘Avrona nature reserve, and two more on 2 January 2019, for the experiments (Appendix C), as well as samples of oil-contaminated and clean soil from the 2014 spill area in the reserve. All females were collected with their complete nests, as each burrow was excavated carefully. The small number of individuals removed for the experiments was due to the area being a sensitive nature reserve and only minimal collecting is permitted. Prior to the experiment, spiders were kept individually in plastic boxes with shredded cardboard and a cardboard cylinder that simulated the spider’s burrow (Figure 6A). The room was maintained at 24 ± 1 °C and light conditions similar to outdoors. Spiders were fed once a week with one ant: *Messor hebraeus* (Santschi, 1927) or *Camponotus sanctus* (Forel, 1904) (Figure 6B). Prior to the experiment, the cephalothorax length of the spiders was measured to estimate their size. Two series of soil preference experiments were conducted: a five-day experiment and a 24-h experiment. All experiments were conducted by placing each spider in an individual arena. The arenas were 20.5 cm diameter round aluminum trays (Figure 6C,D). Half of the arena was filled with clean soil (control) and the other half with oil-contaminated soil to a depth of about 3 cm. Two artificial burrows, 1 cm depth, and 1 cm diameter, were dug 7 cm from the center of the arena and 7 cm from each other. The artificial burrows were lined with paper to prevent the sand from collapsing. The center line of each arena faced North, and all arenas were covered with fine mesh to prevent the spiders from escaping. The arenas were placed in a darkened room maintained at 25 ± 1 °C with only a dim light penetrating the room from outside. The location in the arena of each spider was recorded at the beginning of the experiment and at its end.

#### 2.3.1. Five-Days Experiment

The spiders’ soil preference was tested by placing each spider in a Petri dish (diameter 3.5 cm) level with the soil surface in the center of the arena. Each of the ten spiders was tested twice, with tests separated by three days. Each spider was left in the arena for five days in each trial. The location of the spider in the arena was recorded on the first night, the last night of the experiment, and the last day (right at the end of the experiment).

#### 2.3.2. 24-h Experiment

The effect of the spiders’ initial location on its soil preference was tested to determine whether the type of soil the spider was exposed to at the beginning of the experiment affects the later soil preference of the spider. Half of the spiders were released on the control side of the arena at the start of the experiment and the other half on the oil-contaminated soil side. In this experiment, each spider was tested four times, with two consecutive trials out of four, separated by five days, and with the treatment type of the spiders (the releasing side) reversed at every trial (Appendix C). The location of the spider in the arena was recorded after 24 h at the end of the experiment.

### 2.4. Sahastata sp. nov. Description and Natural History

The *Sahastata* we found resembled the species *Sahastata nigra* (Simon, 1897) that was reported previously from Egypt [28,33]. However, a recent revision of the genus clarified the taxonomic limits of the species *Sahastata nigra* [28]. We therefore described the *Sahastata* we found as a new species to science. All material preserved in 75% ethanol and deposited at the Israel National Arachnid Collection, National Natural History collections, The Hebrew University of Jerusalem (NNHC, HUJI), Steinhardt Museum of Natural History, Tel-Aviv University (SMNH), Museo Argentino de Ciencias Naturales “Bernardino Rivadavia”, Buenos Aires, Argentina (MACN), Forschungsinstitut und Natur-Museum Senckenberg (SMF), and Musee National d’Histoire Naturelle, Paris (MHNH). Transliterated names of the localities follow the “Toponomasticon. Geographical gazetteer of Israel” published by Survey of Israel (1994) and “Israel Touring Map” (1:250,000). Geographic coordinates are given in WGS84. Morphological descriptions and specimen lists were generated following Magalhaes [34]. Morphological examination of the specimens was performed using a Nikon SMZ25 motorized stereomicroscope mounted with a Nikon DS-F12 camera driven by NIS-Elements software 5.21.03 64-bit (NNHC, HUJI), Nikon AZ100 zoom stereomicroscope, and Nikon Eclipse 80i stereomicroscope mounted with Nikon DS-2MBW camera driven by Nikon DS-L2 control unit (Chipman lab, HUJI), Olympus BH-2 compound microscope (MACN), and Olympus SZX16 stereomicroscope mounted with a Canon EOS 7D camera (SMNH). Image stacks were combined using Zerene Stacker (ver. 1.04, Richland, WA, USA) and Helicon Focus (ver. 7.6.2 Pro, Kharkiv, Ukraine) and edited using GIMP (ver. 2.10.10, https://www.gimp.org/ (accessed on 21 July 2021)) and Inkscape (ver. 0.92.4, https://inkscape.org/ (accessed on 21 July 2021)). 10% KOH solution or a pancreatin solution were used for clearing the female genitalia. All measurements are given in millimeters. Leg measurements were taken from the dorsal side, measurements of legs are given in the order of femur, patella, tibia, metatarsus, tarsus.

#### *Sahastata* Burrows

The spiders were found in nature in silk-lined nests in burrows in the ground, and it was unclear whether they dig the burrows themselves or use burrows constructed by other organisms. To clarify this, we placed ten adult females of *Sahastata* in a container with clean soil collected from ‘Avrona nature reserve. Half of the containers included an artificial burrow, and the other half did not. We investigated the web and burrow-building behavior of the spiders and documented whether adult female *Sahastata* build their own burrows or use artificial ones. *Sahastata* females keep the consumed prey inside the burrow; to determine the spider’s diet, we collected the contents of nests of females that were removed from the field for the soil preference experiments and counted and identified the prey remains.

### 2.5. Statistical Analyses

#### 2.5.1. Monitoring—Spatial Dispersion

We calculated the spatial dispersion patterns of the burrows in plots in the clean soil (control) and oil-contaminated areas of 2014 and 1975 spills, using the mean and variance of the number of burrows in each plot type (16 plots total, 4 in each plot type). We calculated the sample mean as the sum of all burrows divided by the total number of plots of each type. To calculate the variance for each plot type, we summed the squared differences between the number of burrows observed in each plot minus the overall mean number of burrows for the plot type. This value was divided by the number of plots minus 1. We then used the ratio of the variance to the mean to determine whether the pattern was uniform or clumped (dispersion index) [35]. A ratio greater than one indicates that individuals are clumped, and a ratio less than one indicates a uniform distribution. A ratio equal to one indicates individuals randomly distributed in space. We used binomial z-ratio to test the burrow distribution between the different plots.

#### 2.5.2. Quadrat Survey, Nearest Neighbor and Habitat Characteristics

We calculated the percentage cover of stones in the quadrats using the photographs taken in the field. The photographs were cropped and rotated with Adobe Photoshop cs6 version 13.0 × 32 (San Jose, CA, USA). The relative area of stones in the images was determined with *ImageJ 1.5* [36], using the plugin *Trainable Weka Segmentation* [37]. We used χ^2^ and one-way ANOVA to test whether soil type, percentage cover of stones, and soil hardness differ between the *Sahastata* burrow and no-burrow quadrats, as well as control and oil-contaminated areas of 2014 and 1975 spills. Moreover, we tested whether the distance of the *Sahastata* burrow, or the center of the no-burrow quadrat, to the nearest *Sahastata* burrow, shrub, tree, and ant nest differ between the *Sahastata* burrow and no-burrow quadrats, as well as control and oil-contaminated areas of 2014 and 1975 spills.

#### 2.5.3. Soil Preference: Laboratory Experiments

χ^2^ test of goodness of fit was performed to determine whether significantly more spiders were found in the clean soil (control) than in the oil-contaminated soil side of the arena. We used logistic regression (with logit link function) to test the effect of the initial location of the spider as the main factor, and the spider identity and trial number as random factors, on the final location of the spider in the arena.

## 3. Results

### 3.1. Monitoring

We found 139 *Sahastata* burrows during the five years of monitoring (May 2016–August 2020) at the 2014 and 1975 spill areas in ‘Avrona nature reserve. Significantly more burrows were found in the clean soil (control) plots in both the 1975 oil-spill area and in the 2014 oil-spill area (Figure 7A). In the 1975 area, there were 38 burrows in the oil-contaminated plots and 55 in the clean plots (binomial z-ratio= −1.66, *p* = 0.0485); in the 2014 site, there were four burrows in the oil-contaminated areas and 42 in the clean soil (control) plots (binomial z-ratio = −5.46, *p* < 0.001). The few burrows that were found in the 2014 oil-contaminated plots, were found near the oil-contaminated soil but never directly in it. Variance to mean ratios indicated a highly clumped distribution of *Sahastata* burrows: in both the 2014 plots (3.933, *n* = 12) and in the 1975 plots (2.47, *n* = 19). Burrows were clumped in both the oil-contaminated and clean soil (control) plots of the 1975 oil-spill area (variance/mean, 2.05, *n* = 11 and 2.99, *n* = 8, respectively). Too few burrows were found in the 2014 oil-contaminated plots to assess the distribution. Of 139 burrows, 64 (46%) were documented with a spider at least once and another 14 (10%) had fresh silk, but the spider was not detected. Sixty-one burrows (ca. 44%) had no documentation of a spider and seemed abandoned or inactive. These were assigned as unknown or no spider (Figure 7B). Fifty-nine burrows (of 126, after removing the burrows of the collected spiders) were documented only once, and 67 were seen on one or more subsequent visits. Sixty-two burrows had adult or subadult females, and only five were documented with juvenile individuals, two of which recolonized burrows previously occupied by adult females. During the five-year monitoring, fourteen *Sahastata* spiders were found in two consecutive years, four in three consecutive years, and three in four consecutive years.

### 3.2. Quadrat Survey, Nearest Neighbor and Habitat Characteristics

Spiders were present or silk nests were fresh in all 18 *Sahastata* burrows in the quadrat survey. The distance between each *Sahastata* burrow, or quadrat center for non-burrow quadrats, to the nearest *Sahastata* burrow was not significantly different between oil-contaminated soil and clean soil quadrats in either 1975 or 2014 areas. The distance between each *Sahastata* burrow, or quadrat center for non-burrow quadrats, to the nearest shrub was not significantly different between quadrats with or without a burrow, but distances were shorter in oil-contaminated soil compared to clean soil (χ^2^ = 4.6, df = 1, *p* = 0.03). Soil type, stone cover, and hardness did not differ significantly between burrow and non-burrow quadrats within each of the areas (1975 and 2014), but soil hardness and stone cover were very heterogeneous within and between quadrats. The height of the silk, projecting out of the burrow, ranged between zero (ground level) to three centimeters, with no significant differences in burrow height between 1975 and 2014 or oil-contaminated soil and clean soil quadrats. Burrow opening diameter ranged between 0.3–1.7 cm, with no significant differences between 1975 and 2014 quadrats, but burrow opening diameter was significantly smaller in the oil-contaminated soil quadrats compared to the clean soil quadrats (mean diameter in oil-contaminated soil = 0.76 cm, mean diameter in clean soil = 1.2 cm, one-way ANOVA: F = 6.6, df = 1, *p* = 0.02) (Figure 7C).

### 3.3. Soil Preference: Laboratory Experiments

The cephalothorax length of the ten spiders used for the experiments ranged between 4.2 mm to 5.5 mm, and their maximum original burrow opening length ranged between 1.2 cm to 3 cm, with no significant correlation between the cephalothorax and original burrow sizes (correlation coefficient = −0.59, *p* = 0.07; Appendix C). In all trials, spiders were active during the night and did not move much during the daytime.

#### 3.3.1. Five-Days Experiment

Three individuals added silk around and in the artificial burrows in the clean soil side of the arena (control), two in the oil-contaminated soil side, one in both, and four did not add silk on the soil. On the last day of the experiment, significantly more spiders were found on the clean soil (control) side of the arena (17 out of 20; χ^2^: χ^2^ = 13.056, *p* = 0.0015). On the first and last night there was a trend of more spiders (Figure 8A), however, this was not statistically significant, as some spiders occupied the border area (χ^2^: First night χ^2^ = 4.222, *p* = 0.1211; Last night χ^2^ = 5.056, *p* = 0.0798).

#### 3.3.2. 24-h Experiment

In this experiment, we tested if there is an effect of the initial location at the beginning of the experiment (clean soil versus oil-contaminated soil), spider identity, and trail on the spider’s soil preference after 24 h. We found more spiders in the clean soil (control) side of the arena at the end of the experiments (combined trials, 27 out of total 40 observations, Figure 8B), suggesting a significant preference for clean soil and no significant effect of the spider identity, initial location, and trial number (logistic regression with effect likelihood ratio tests, spider identity: χ^2^ = 8.218, *p* = 0.512; initial location: χ^2^ = 0.192, *p* = 0.661; trial: χ^2^ = 1.633, *p* = 0.652).

### 3.4. Natural History and Description

#### 3.4.1. Taxonomy and Description

Family Filistatidae Simon, 1864.

Genus *Sahastata* Benoit, 1968.

*Sahastata* Benoit, 1968: 96 [38]; Patel 1978: 186 [39]; Brignoli 1982: 74 [40]; Marusik et al., 2014: 5 [41]; Marusik & Zamani 2015: 125 [42]; Marusik & Zamani 2016: 268 [43]; Zonstein & Marusik 2019: 90 [44]; Magalhaes et al., 2020: 220 [28].

Type species. *Filistata nigra* Simon, 1897: 97. Lectotype female (MHNH AR 14851) and 4 female paralectotypes (MNHN AR 5476): OMAN: Muscat, X.1896, leg. M. Maindron—examined. For a description of the genus and redescription of the type species see Magalhaes et al. (2020).

***Sahastata aravaensis*****sp. nov.** Ganem, Magalhaes, Zonstein and Gavish-Regev**. Figure 9, Figure 10, Figure 11, Figure 12 and Figure 13**. LSID: urn:lsid:zoobank.org:act:C11F3391-DFC6-41F9-94FB-F0B0DD58D9B0.

**Holotype: Israel.** ‘**Arava valley:** ‘Avrona Nature Reserve (29.67462° N, 35.00213° E, 36 m.a.s.l.), hyper-arid desert, burrow in soil with loess, sand and gypsum, hand collected, Efrat Gavish-Regev, Yebin Byun, Assaf Uzan & Ibrahim Salman coll., 28 November 2018, 1 ♀ (HUJI-Ar 20302).

**Paratypes: Israel.** ‘**Arava valley:** ‘Avrona Nature Reserve (29.67008° N, 34.99575° E, 43 m.a.s.l.), hyper-arid desert, burrow in gravel, soil with loess, hand collected, Efrat Gavish-Regev, Igor Armiach Steinpress, Ibrahim Salman, Marjia Majer, Yael Lubin coll., 6 November 2016, 1 ♀ (HUJI-AR 20303), same locality (29.67878° N, 35.00038° E, 42 m.a.s.l.), Efrat Gavish-Regev, Igor Armiach Steinpress, Ibrahim Salman, Marjia Majer, Yael Lubin coll., 7 November 2016, 1 ♀ (HUJI-AR 20305), same locality (29.67884° N, 35.00042° E, 42 m.a.s.l.), hand collected with juveniles, same collector and date, 1 ♀ (HUJI-AR 20304), same locality (29.67949° N, 34.99946° E, 44 m.a.s.l.), Efrat Gavish-Regev, Yebin Byun, Assaf Uzan & Ibrahim Salman coll., 28 November 2018, 1 ♀ (MACN-Ar 41837), same locality (29.68094° N, 34.99762° E, 46 m.a.s.l.), hyper-arid desert, burrow in soil with loess, sand and stones, hand collected, Efrat Gavish-Regev, Yebin Byun, Assaf Uzan & Ibrahim Salman coll., same date, 1 ♀ (HUJI-AR 21001), south of ‘Avrona Nature Reserve, near the salt pools (29.62694° N, 34.99006° E, 31 m.a.s.l.), hyper-arid desert, burrow in gravel from Timna’ for building, hand collected, Nitzan Segev coll., 14 April 2019, 1 ♀ (HUJI-AR 21002), Nahal Shezaf (30.72424° N, 35.27176° E, 125 m.b.s.l.), hyper-arid desert, burrows in sandy soil on bank terrace, hand collected, Sergei Zonstein coll., 29.III.2011, 2♀ (SMNH), Yotvata sand dune (29.88903° N, 35.07663° E, 73 m.a.s.l.), hyper-arid desert, burrow in sand dune, hand collected, Nitzan Segev coll., 20 November 2021, 1 ♀ (HUJI-AR 21003). **Jordan.** No further data (label illegible), R. Kinzelbach coll., J. Wunderlich vend. 2008, *n*. 35, 1 ♀ (SMF).

**Additional material examined: Israel. ‘Arava valley:** Shezaf reserve near Hazeva, hyper-arid desert, Ori Segev coll., 9.II.1998, 1♀ (HUJI-AR 21000), Timna’, hyper-arid desert, Ascher coll., 21 April 1957, 1♀ (HUJI-AR 21005), Elat, hyper-arid desert, Benny Shalmon coll., 1 November 1971, 1♀ (HUJI-AR 21004).

**Etymology**. The specific name refers to the type locality.

**Distribution**. The ‘Arava desert in Israel; Additional burrows were photographed by Nizan Segev from *Hoodia parviflora* N.E. Br. cultivation near Yotvata on December 19, 2014 (29.88° N, 35.07° E), from Shahaq wadi (30.78° N, 35.27° E) on November 5, 2015, from north of Lotan (30.033614° N, 35.096799° E) on December 10, 2020, from the central ‘Arava, and by Liri Kopelvich from Shahaq wadi south to Hazeva (30.758308° N, 35.258843° E) on December 15, 2021; also recorded from an uncertain locality in Jordan. A female from Yemen previously identified tentatively as *S. infuscata* (Kulczyński, 1901) by Magalhaes et al. (2020, Figure 25C, [28]) has a similar spermatheca with a tapering membranous portion and might be conspecific or closely related to *S. aravaensis.*

**Diagnosis**. Females are similar to those of *S. infuscata*, *S. nigra* and *S. wesolowskae* Magalhaes, Stockmann, Marusick & Zonstein by possessing two small spermathecae with a glandular portion embracing an elongate membranous portion (Figure 9 and Figure 10). Females of *S. aravaensis* differ from these species by the broader spaced spermathecae (by about 0.3 mm; closely spaced in *S. infuscata* ); by the kidney shape of the glandular portion of the spermathecae (shorter and lobed in *S. nigra* and *S. wesolowskae*) and by the shape of the membranous portion of the spermatheca: relatively long, tongue-like, and distally tapering with a pointed tip (sausage-like in *S. infuscata* , shorter in *S. nigra* and *S. wesolowskae*) (Figure 9 and Figure 10).

**Description**. Female holotype (HUJI-AR 20302). **Coloration.** Carapace brown, suffused with randomly distributed dark brown lines and with dark brown V-shaped median pattern (Figure 11 and Figure 12G). Labium, endites and sternum dark brown. Sternum and legs I–IV hirsute, with long, thick black setae and with a hairless stripe in the middle of the coxa (Figure 12H). Legs dark brown, with longitudinal brown stripes, gradually becoming brown on the tips. Abdomen dark gray, brown, covered in black setae (Figure 11). Spinnerets uniformly clayish yellow. **Structure and measurements.** Anterior margin of the carapace unmodified. Eye apodemes absent. Sternum suboval, with two pairs of sigillae. Posterior median spinnerets laterally elongate, with one paracribellar gland spigot apparently laterally displaced (Figure 12A–D). Calamistrum with 3 rows of tightly packed and neatly arranged setae, rows with ~23-28-13 setae, the retrolateral one very reduced (Figure 12E,F). Spermathecae with elongate membranous portion distally tapering and embraced by the kidney shaped glandular portion (Figure 9 and Figure 10). Total length 12.28. Carapace length 4.95, width 3.82. Clypeus length 0.88. Eye diameters and interdistances: AME 0.17, PME 0.21, ALE 0.28, PLE 0.22, AME-AME 0.10, PME-PME 0.28. Sternum length 2.28, width 2.17, length with labium 3.47. Palp: femur length 2.34, height 0.92, tibia length 1.49, height 0.75. Leg I: femur 5.46, patella 1.63, tibia 5.17, metatarsus 4.82, tarsus 2.95, total 20.01. II: femur 4.31, patella 1.54, tibia 3.32, metatarsus 3.26, tarsus 2.02, total 14.45. III: femur 3.22, patella 1.33, tibia 2.57, metatarsus 2.72, tarsus 1.67, total 11.51. IV: femur 4.78, patella 1.73, tibia 3.96, metatarsus 3.93, tarsus 2.14, total 16.54. Abdomen: length 7.56, width 4.70. **State of specimen**. Good, right and left legs I-II disarticulated from femur (right leg II was lost before collected), genitalia dissected.

The male is unknown.

**Variation** (based on the holotype and six female paratypes; mean value between parentheses): Total length 9.55–12.45 (10.91). Carapace length 4.37–5.21 (4.76), width 3.4–4.09 (3.76). Clypeus length 0.62–1.05 (0.89). Eye diameters and interdistances: AME 0.16–0.25 (0.21), PME 0.18–0.23 (0.21), ALE 0.24–0.31 (0.26), PLE 0.22–0.25 (0.24), AME-AME 0.07–0.1 (0.1), PME-PME 0.24–0.38 (0.3). Sternum length 1.48–2.58 (2.09), width 2.02–2.48 (2.23), length with labium 3.14–3.82 (3.46). Palp: femur length 2.13–2.66 (2.34), height 0.87–1.11 (0.94), tibia length 1.3–1.62 (1.46), height 0.7–0.99 (0.85). Leg I: femur 4.91–5.93 (5.48), patella 1.36–2.04 (1.62), tibia 4.8–5.51 (5.08), metatarsus 3.76–5.01 (4.53), tarsus 2.08–3.06 (2.52), total 17.5–21.23 (19.22). II: femur 3.67–4.41 (4.09), patella 1.28–1.88 (1.58), tibia 3.11–3.49 (3.29), metatarsus 2.87–3.31 (3.13), tarsus 1.8–2.07 (1.95), total 12.8–14.68 (14.04). III: femur 2.89–3.46 (3.16), patella 1.29–1.58 (1.41), tibia 2.18–2.64 (2.39), metatarsus 2.31–2.77 (2.59), tarsus 1.56–1.77 (1.65), total 10.27–11.94 (11.2). IV: femur 4.4–4.89 (4.67), patella 1.36–1.87 (1.7), tibia 3.14–4.26 (3.74), metatarsus 2.96–3.95 (3.59), tarsus 1.89–2.2 (2.06), total 14.13–16.8 (15.75). Abdomen: length 5.36–7.56 (6.21), width 2.86–4.8 (3.83).

#### 3.4.2. Natural History

Females are found in the hyper-arid desert, in a variety of soil types such as sand and loess, sometimes covered by gravel, in soil with gypsum and near wadies (dry riverbeds) dominated by desert vegetation such as Acacia trees and perennial shrubs (Figure 2). Nests were probably constructed inside pre-existing burrows, usually away from the vegetation (Figure 3 and Figure 4), similar to other congeners (Magalhaes et al., 2020, Figure 2 [28]). All the burrows we excavated (a total of 14), were 7–15 cm in length (Figure 13). In these burrows, usually, the upper part of the burrow was vertical to the soil surface, followed by some sort of bending. In some of the burrows after the bending, the burrow becomes horizontal and parallel to the ground level, with a layer of gypsum separating the burrow from the surface. Of the 14 spiders and burrows collected, three females had 1–5 egg-sacs and one or a few juveniles within their burrows. No males were documented, but in many spider families, males would be caught only in pitfall traps (during mate search). As part of the monitoring, we used pitfall traps, but only for two weeks (in early summer and late summer), and we did not collect any male *Sahastata*. In addition, it was suggested for other *Sahastata* species that males are short-lived after maturity, and in captivity, they account for 20–25% of the juveniles raised [28]. We kept female spiders in captivity in the laboratory for two to three years and fed them mainly with ants. The spiders usually bite the ant’s leg before feeding (Appendix A).

#### 3.4.3. Prey from Collected Nests

In six silk nests of the burrows of both spill areas (control 2014: 3; control 1975: 2; oil 1975: 1), remains of ants, mainly *Cataglyphis nigra* (André, 1881) *Camponotus* Mayr 1861 sp. and *Monomorium* Mayr 1855 sp. (Formicidae) (37.5–100%) and of darkling beetles (Coleoptera, Tenebrionidae) and other beetles (0–62.5%), were found together with other insects such as Lepidoptera larvae (0–9%) and Apoidea (0–18%). In addition, in the 1975 burrows, remains of Araneae (6%) were found, while in the burrows of 2014, remains of Diptera (9%) and Solifugae (3%) were found. Results from pitfall-trap monitoring suggest that ants comprise ~90% of the arthropod assemblage at ‘Avrona nature reserve (Nitzan Segev, personal communication). As *Sahstata aravaensis* sp. nov. is a sessile predator, and *Cataglyphis* ants are very active, it may be that the high proportion of those ants in the prey remains is due to their dominance in this habitat.

#### 3.4.4. Burrow Use in Laboratory

All five females that were placed in a container with an artificial burrow constructed silk nests within one week in the artificial burrow. None of the five females that were provided with clean soil in the plastic boxes without artificial burrow dug a burrow. Instead, they spun their webs on the entire surface of the container.

#### 3.4.5. Relationships

DNA sequence data obtained from *Sahastata aravaensis* (refer to as *Sahastata* indet. (IFM-0269, Figure 4) in Maglhaes et al., 2020 [28]) indicates this species is more closely related to *Sahastata nigra* and *Sahastata wesolowskae* (both from Oman) than to *S. wunderlichi* (from Morocco) [28] which is consistent with the similarities observed in the morphology of the genitalia and posterior median spinnerets.

## 4. Discussion

In the present study, we found that a sedentary desert burrow-dwelling spider, *Sahastata aravaensis* sp. nov., was severely negatively affected by two oil spills that occurred in a hyper-arid desert in the ‘Arava valley. Our five-year monitoring suggests a direct reduction in burrows occupied by these spiders due to the 2014 oil spill in the ‘Avrona Nature Reserve, with only four burrows found in the oil-contaminated soil plots during the entire monitoring, compared to 42 burrows in the adjacent clean soil plots (Figure 7A). The 2014 oil spill was an after-the-event investigation and therefore lacks a ‘before’ control [29] (this is also correct for the 1975 oil spill). Nevertheless, even after 40 years, the number of burrows in the oil-contaminated plots of the 1975 spill was significantly less than in uncontaminated clean soil plots. This further supports the hypothesis that damage to the habitat due to the oil spills caused a long-term reduction in *S. aravaensis* abundance. It is also possible that *S. aravaensis* is less abundant in the oil-contaminated areas due to disturbance by naturally occurring floods rather than the oil flow. However, flooding could affect both oil-contaminated and clean soil areas. We found in the 2014 area three burrows in the oil-contaminated plots and five in the clean soil plots that were destroyed by floods; two of the burrows in the clean soil were reconstructed after the flood. The wadis in both areas are shallow and floodwaters do not necessarily follow the trajectory of the oil spill. Thus, flooding is unlikely to solely explain the significant difference in burrow numbers in oil-contaminated and clean soils. Manipulations following the 2014 oil spill might also have had an effect on spider survival. We did not specifically test the effects of soil tillage and treatments to stimulate oil biodegradation [15], but we suggest that these may pose an additional threat to these environmentally sensitive burrow-dwelling spiders.

In the laboratory experiments, spiders showed a clear preference for the clean soil side of the arena over the side with oil-contaminated soil. Initial exposure to clean vs. contaminated soil did not influence this preference. The spiders were removed from the field where they were found in uncontaminated clean soil, so they had no previous experience. However, the same spiders were tested more than once (due to restrictions on collecting in the nature reserve; see Methods), so it is possible that an initial aversion to the contamination was reinforced over the course of the experiments. It is not clear if the avoidance of the oil-contaminated soil was due to toxicity upon contact to a repellent effect of volatile compounds in the oil-contaminated soil, or the dark color of the soil. The cuticle of some species of spiders that have dense hairs covering the body was shown to have high crude oil-absorbent properties [45]; this characteristic may increase the vulnerability to oil spills of species such as *S*. *aravaensis*.

We hypothesized that digging in oil-polluted soil could be more difficult than in clean soil. Morphologically, we could not identify any structure that enables adult females to make their own burrow. None of the five females that were provided with clean soil in the plastic boxes in the laboratory dug a burrow, and all five females that were given an artificial burrow constructed silk nests within one week in the artificial burrow. We suggest, therefore, that *S.*
*aravaensis* uses already-made burrows in the habitat. If this assumption is correct, any manipulation of the soil may ruin also vacant burrows that could be colonized by *Sahastata*. As we observed in our monitoring both small individuals in small burrows with small nests and large individuals in large burrows with large nests, we suspect spiders may enlarge their burrows with time. Less probable, but still possible, is that spiders move into abandoned *Sahastata* nests or other empty burrows such as those of small lizards or empty ant nests.

Burrow opening diameter of *S.*
*aravaensis* was significantly smaller in 1975 oil-contaminated soil quadrats compared to other quadrats (Figure 7C). We suggest that spiders in the oil-contaminated quadrats were unable to expand their burrows due to the nature of the soil in the untreated spill areas. Although we did not find greater soil hardness in the 1975 oil-contaminated soil survey quadrats, the measurements were only of the soil crust, and not of deeper parts of the soil, where digging may be more difficult due to a layer of solidified oil. Other changes in soil characteristics, such as extreme hydrophobicity [10] that resulted from the oil spill might affect the spiders’ ability to expand their burrows. Additional explanations could be a lower availability of larger empty burrows or smaller spider size in the oil-contaminated areas due to lower prey availability. However, burrow diameter and spider size were not correlated (see Results), and the likely main prey of *S.*
*aravaensis*—ants—were significantly more abundant in the 1975 oil-contaminated plots than the clean oil (control) plots [46]. A similar pattern of high ant abundance in oil-contaminated soils was found also in saltmarshes in the USA [47], and in a desert in Kuwait [48].

Finally, the clumped dispersion observed for *S*. *aravaensis* in our monitoring plots, suggests these spiders are slow and short-distance dispersers, which would further slow population recovery after a severe local disturbance. Desert soil surfaces are exposed and young spiders dispersing from the maternal burrow risk predation by numerous arthropod and vertebrate predators, as well as unfavorable weather conditions at the surface. These threats may select for short-range dispersal, a pattern commonly seen in many burrowing spiders [49]. Thus, in addition to possible negative effects of soil contamination by the oil spill, short-range dispersal may prevent the recolonization of extended areas following a disturbance. In the light of our findings, we suggest that *S*. *aravaensis* can serve as a bioindicator for the recovery of the soil habitat after oil spills in this system. Several studies demonstrated the use of burrow-dwelling spiders and active hunters as bioindicators for bioaccumulation of heavy metals [50,51], however many individuals must be sacrificed to quantify chemical accumulation in their tissues. This is a major drawback when it comes to an organism whose conservation status is unknown, as in this research. In our study, we show that burrow-dwelling spiders can be used as bioindicators of soil pollution in the habitat by monitoring their spatial distribution.

## Figures and Tables

**Figure 1 insects-13-00101-f001:**
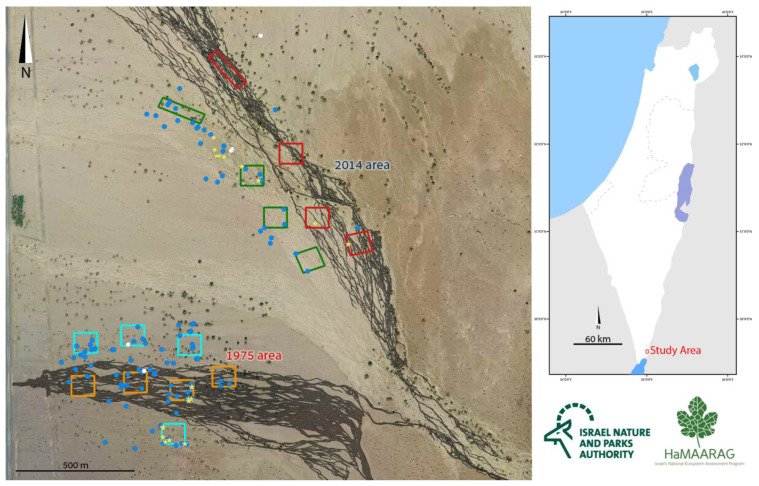
The study area with the 2014 and 1975 oil spill areas are marked in black. Rectangles designate monitoring plots: 2014 oil-contaminated plots in red, 2014 uncontaminated clean soil (control) plots in green, 1975 oil-contaminated plots in orange, 1975 uncontaminated clean soil (control) plots in turquoise. Dots and stars designate *Sahastata* burrows: burrows found in the first year of monitoring (2016) in white, burrows found after 2016 in blue, yellow stars designate burrows used for the quadrate survey.

**Figure 2 insects-13-00101-f002:**
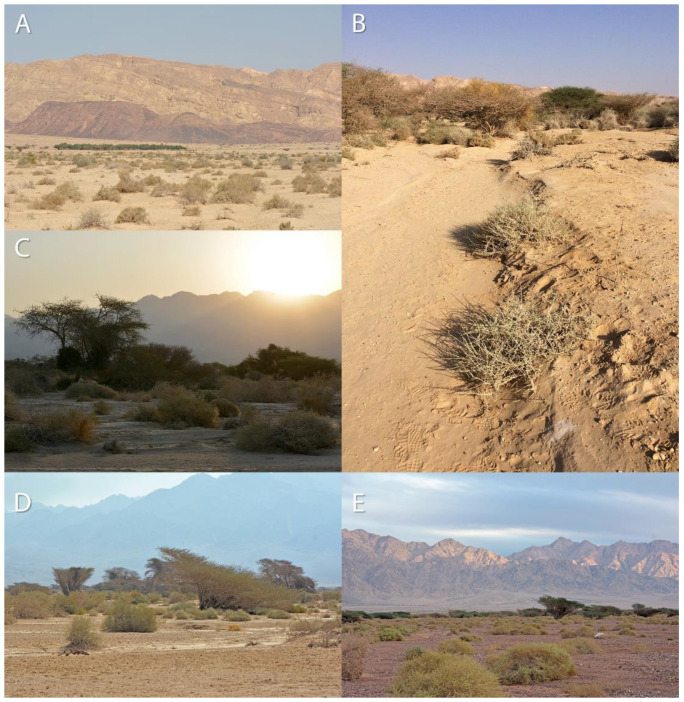
‘Avrona nature reserve habitats. (**A**–**C**) 2014 area dominated by sand and loess soils, (**D**,**E**) 1975 area dominated by loess covered by stones (gravel). Pictures: (**A**,**C**–**E**) Assaf Uzan, (**B**) Ibrahim N. A. Salman.

**Figure 3 insects-13-00101-f003:**
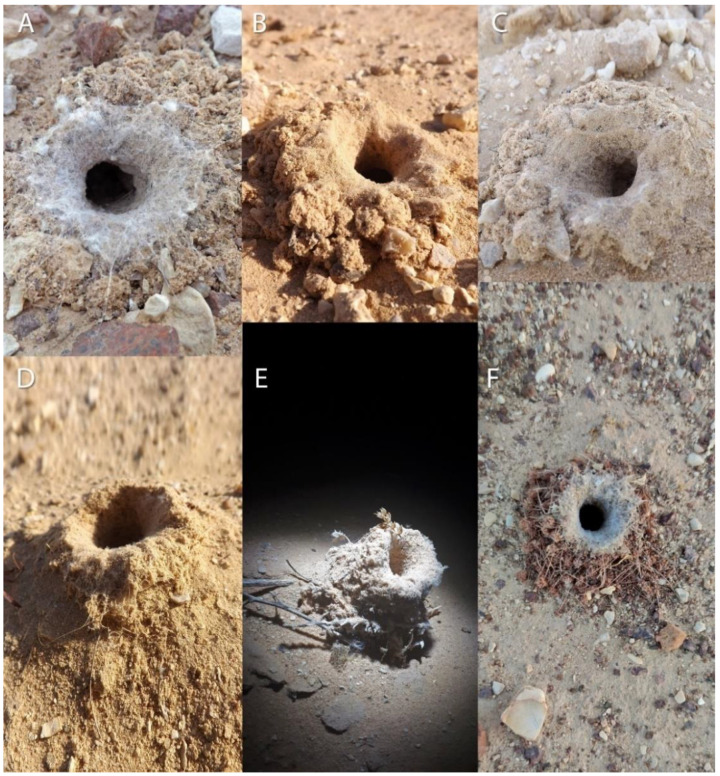
*Sahastata* burrows. (**A**,**C**,**F**): ‘Avrona nature reserve 1975 area dominated by loess covered by stones (gravel), (**B**,**D**): ‘Avrona nature reserve 2014 area dominated by sand and loess soils, (**E**): Yotvata sand dune. Pictures: (**A**–**D**): Igor Armiach Steinpress, (**E**): Nitzan Segev, (**F**): Yoram Zvik.

**Figure 4 insects-13-00101-f004:**
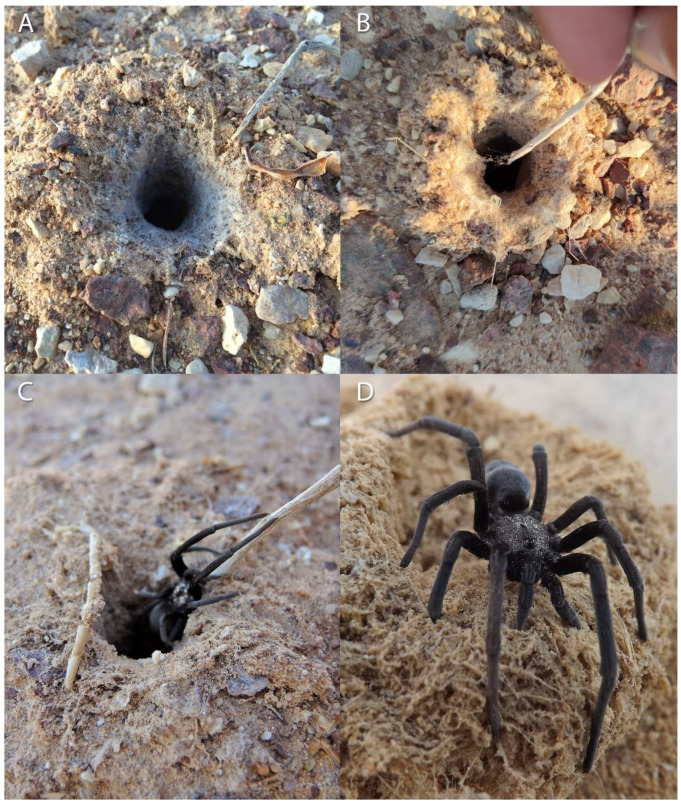
Presence of *Sahastata* spiders in burrows indicated by the spider response to stimulating the silk in the burrow entrance with a twig. Pictures: (**A**): Assaf Uzan, (**B**): Yoram Zvik, (**C**,**D**): Igor Armiach Steinpress.

**Figure 5 insects-13-00101-f005:**
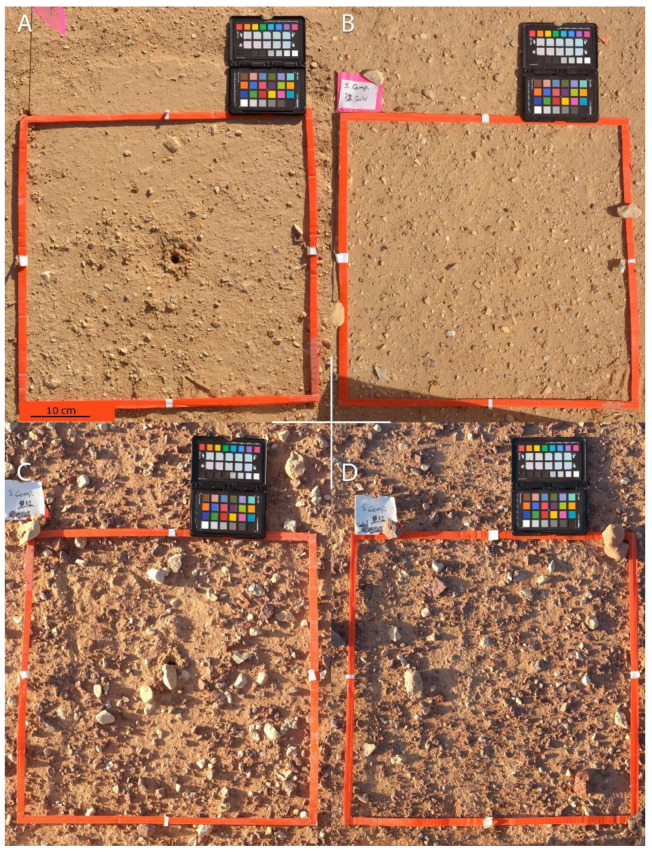
Quadrat survey. (**A**,**B**): Quadrats in ‘Avrona nature reserve 2014 oil-spill area, (**C**,**D**): Quadrats in ‘Avrona nature reserve 1975 oil-spill area. (**A**,**C**): Quadrats with *Sahastata* burrow (18 quadrats), (**B**,**D**) No-burrow quadrats (20 quadrats). Pictures: Assaf Uzan.

**Figure 6 insects-13-00101-f006:**
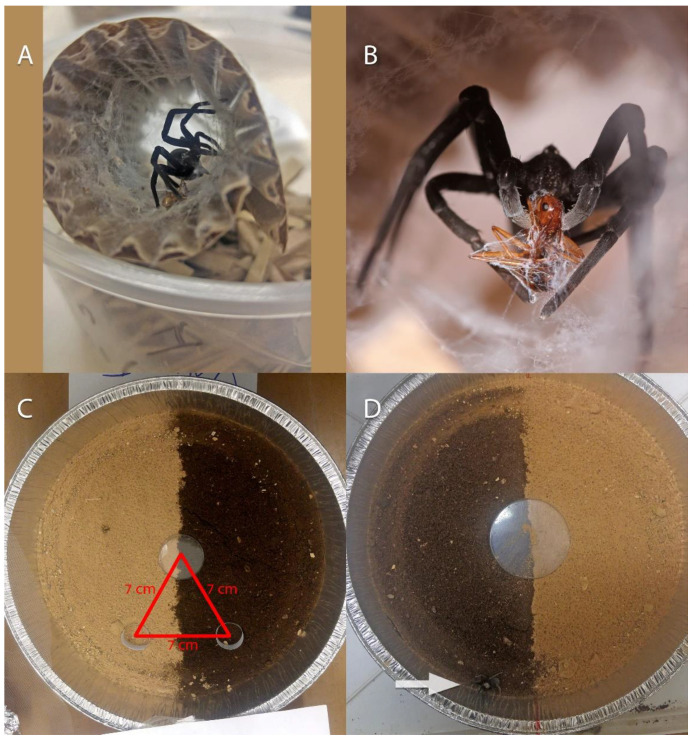
Laboratory spider keeping and experimental setup. (**A**): Plastic boxes with shredded cardboard and a cardboard cylinder used for keeping the spiders, (**B**): Spider feeding on an ant, (**C**,**D**): Arena used for experiments, half of the arena was filled with clean soil (control) and the other half with oil-contaminated soil. Two artificial burrows, 1 cm depth and 1 cm diameter, were dug 7 cm from the center of the arena and 7 cm from each other. The artificial burrows were lined with paper to prevent the sand from collapsing, the arrow (6D) points to the spider location in the arena during one of the replications. Pictures: (**A**,**C**,**D**): Yebin Byun, (**B**): Shlomi Aharon.

**Figure 7 insects-13-00101-f007:**
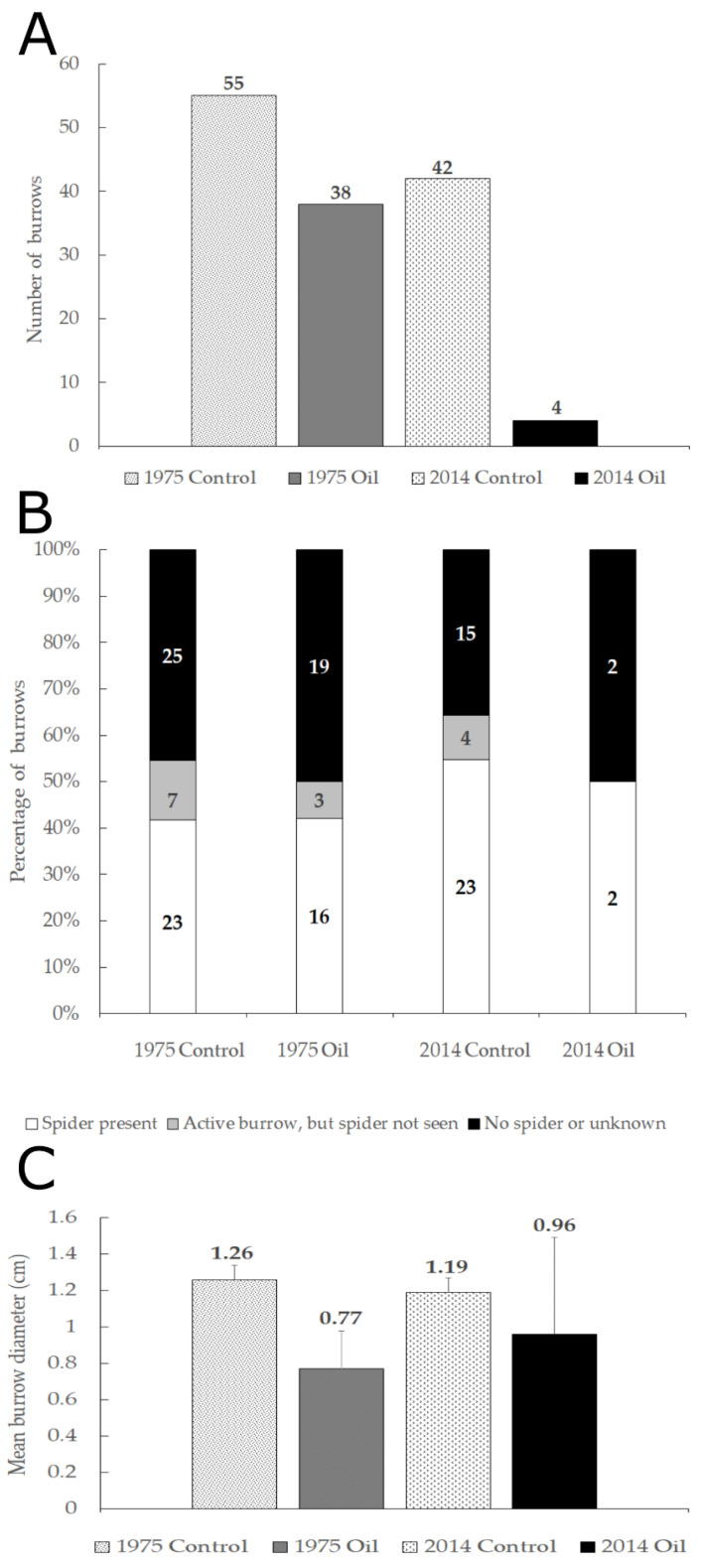
(**A**). Distribution of 139 *Sahastata* burrows observed in four plot types during the entire monitoring period. (**B**). Percentage of burrows with spiders (white), active burrows where spiders were not observed (grey), and with burrows that looks abandoned or where no spider was observed (black) in four plot types during the entire monitoring period. (**C**). Mean burrow opening diameter (centimeters) in the quadrat survey.

**Figure 8 insects-13-00101-f008:**
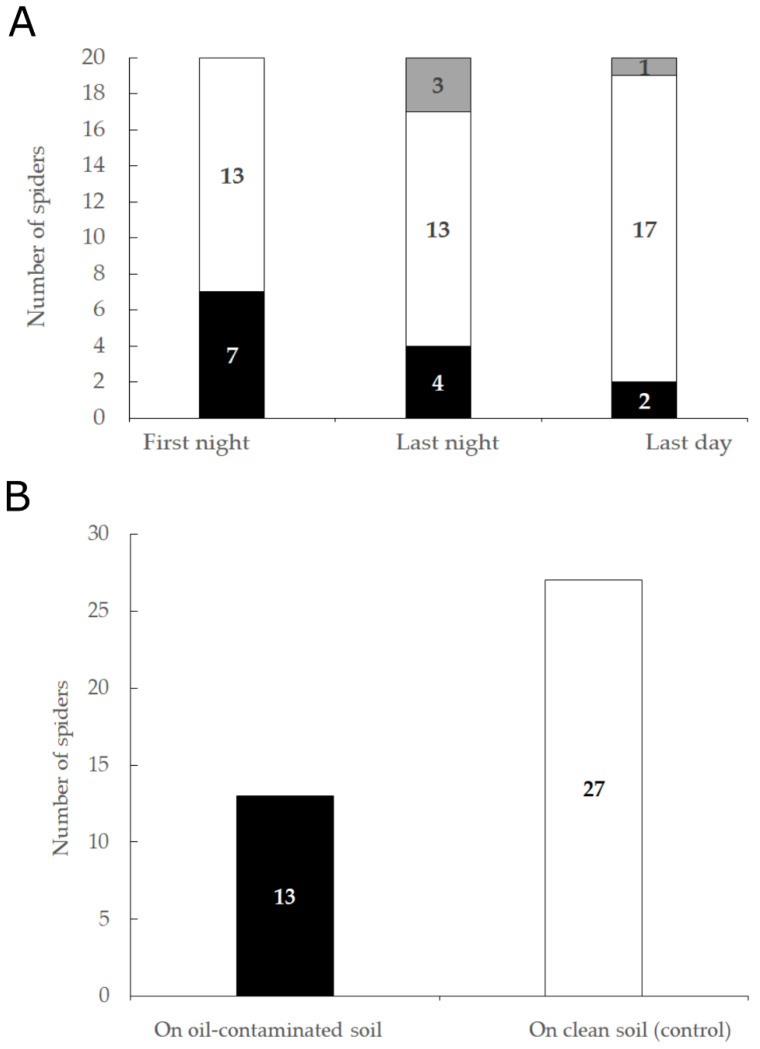
(**A**). The total number of spiders found in each soil type in the arena on the first night, last night, and last day of the five-day experiment. (**B**). The total number of spiders found in each soil type at the end of the 24-h experiment. Black—number of spiders found on the oil-contaminated soil side, white—on the clean soil side (control), and gray at the border between the two sides.

**Figure 9 insects-13-00101-f009:**
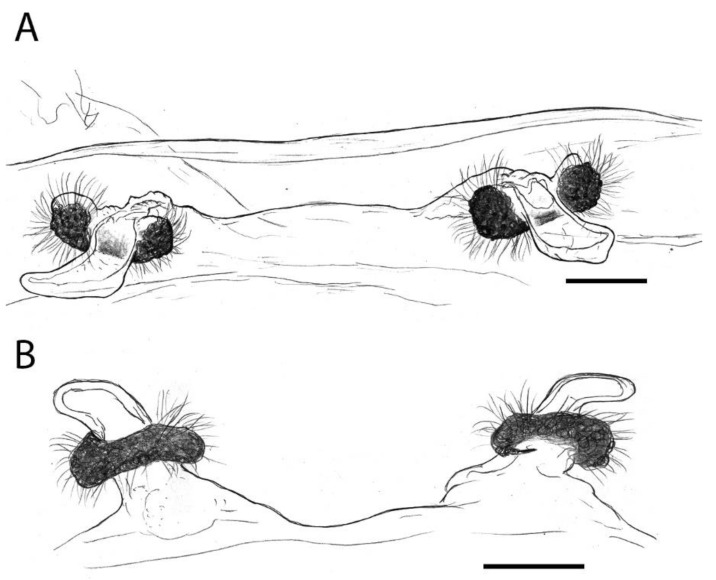
*Sahastata aravaensis* sp. nov. female endogyne. (**A**): paratype MACN-Ar 41837. (**B**): paratype SMF. Drawings: Ivan L. F. Magalhaes, Scale bars = 100 µm.

**Figure 10 insects-13-00101-f010:**
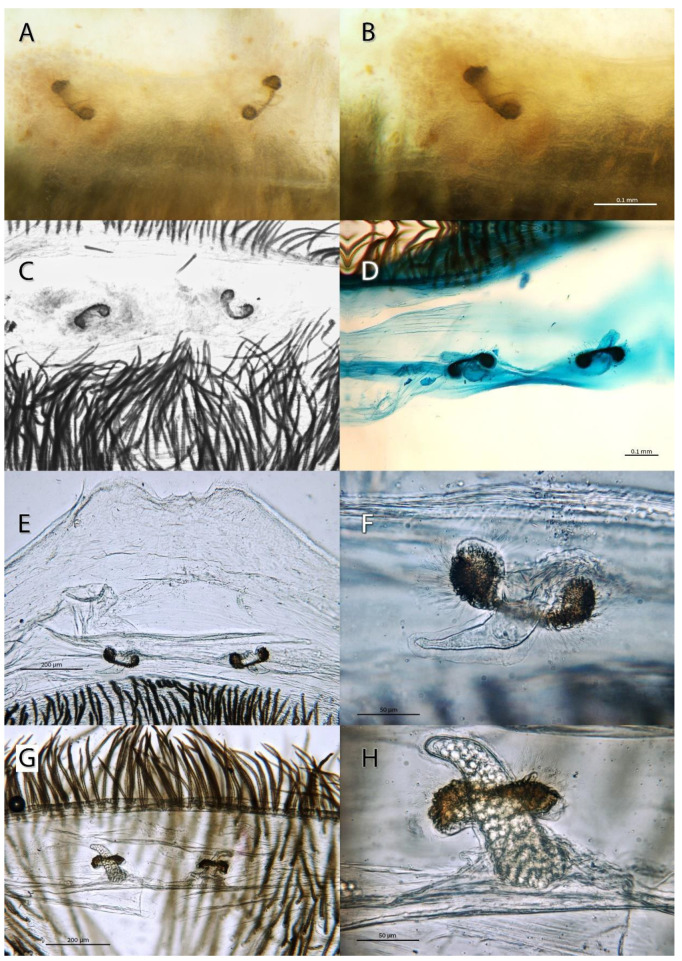
*Sahastata aravaensis* sp. nov. female endogyne. (**A**–**C**): Holotype HUJI-AR 20302, Pictures: Zeana Ganem, (**D**): paratype SMNH, Picture: Sergei Zonstein, (**E**,**F**): paratype MACN-Ar 41837, (**G**,**H**): paratype SMF. Pictures: Ivan L. F. Magalhaes.

**Figure 11 insects-13-00101-f011:**
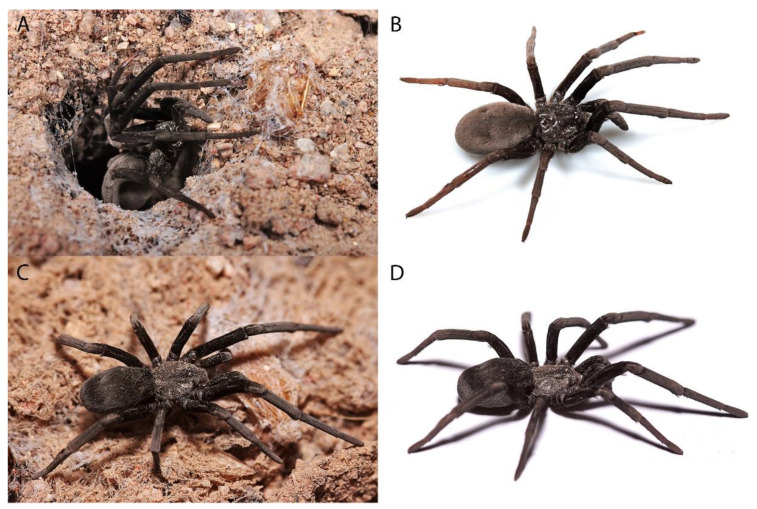
*Sahastata aravaensis* sp. nov. habitus of live spiders in laboratory. (**A**,**B**): paratype MACN-Ar 41837, Pictures: Ivan L. F. Magalhaes, (**C**,**D**): paratype HUJI-AR 21001**,** Pictures: Shlomi Aharon.

**Figure 12 insects-13-00101-f012:**
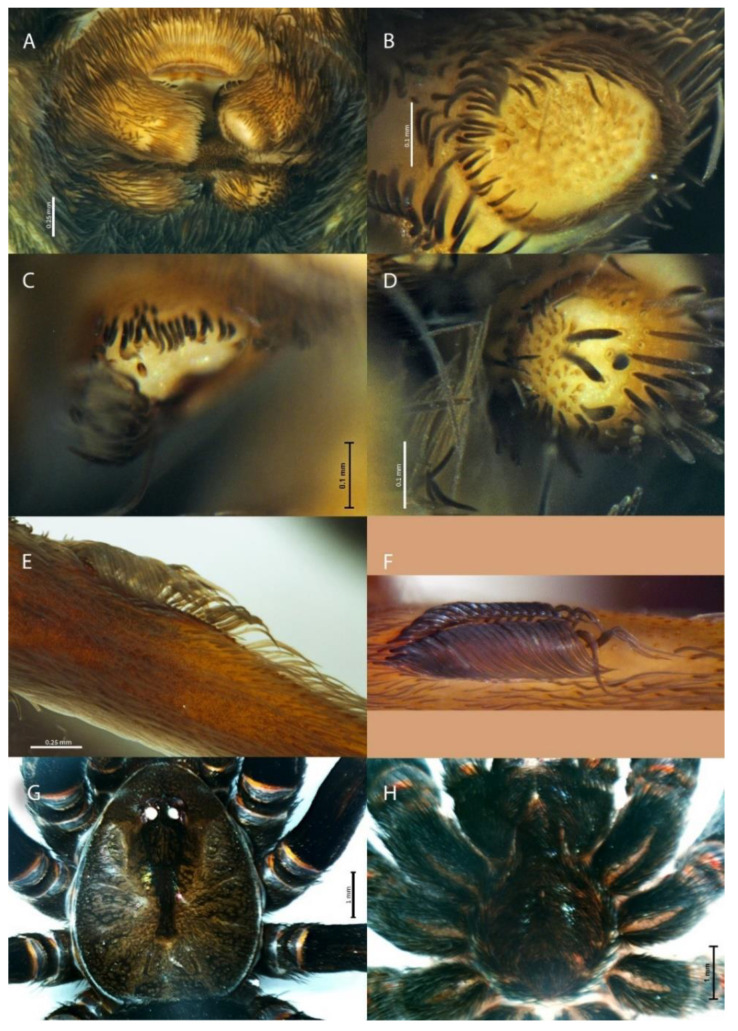
*Sahastata aravaensis* sp. nov. (**A**–**E**): Holotype HUJI-AR 20302, G-C: Paratype HUJI-AR 21001. (**A**): Spinnerets, (**B**): left anterior lateral spinneret, (**C**): left posterior median spinneret, (**D**): left posterior lateral spinneret, (**E**,**F**): calamistrum, (**G**): cephalothorax dorsal, (**H**): sternum. Pictures: (**A**–**E**), (**G**,**H**): Zeana Ganem, (**F**): Efrat Gavish-Regev.

**Figure 13 insects-13-00101-f013:**
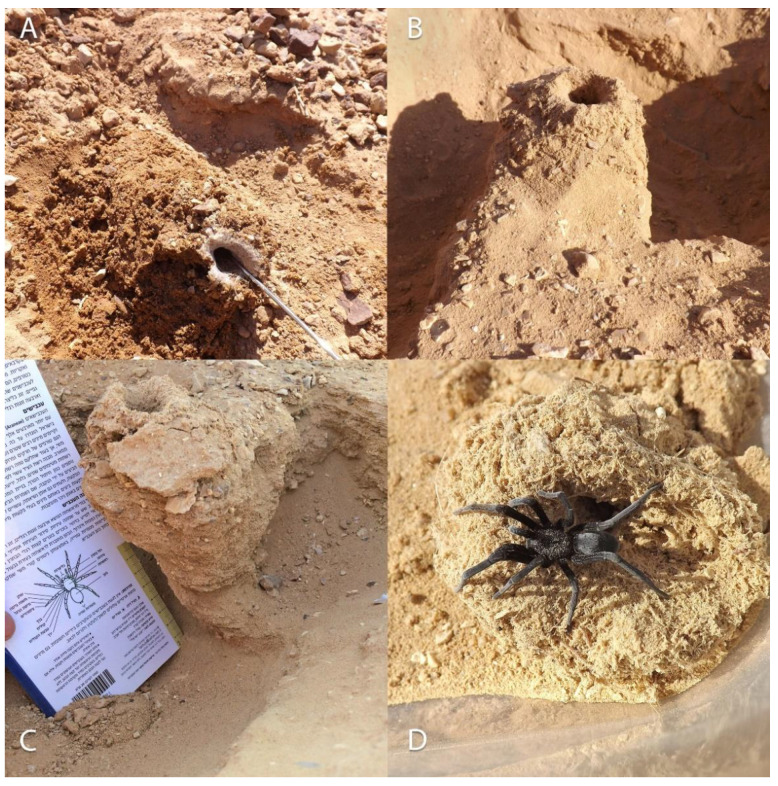
*Sahastata aravaensis* sp. nov. burrow excavated from ‘Avrona nature reserve. Pictures: (**A**,**B**,**D**): Igor Armiach Steinpress, (**C**): Ibrahim N. A. Salman.

## Data Availability

The data from laboratory experiments presented in this study are available in the appendix and the Results section. The monitoring data presented in this study are available on request from the corresponding author.

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
