# Peer review of "Five-Year Monitoring of a Desert Burrow-Dwelling Spider Following an Environmental Disaster Indicates Long-Term Impacts"

_insects, 2022, doi:10.3390/insects13010101_

Round 1
Reviewer 1 Report
This study is an important contribution towards understanding the impacts of oil pollution, and also adds rarely recorded information on an unusual spider family/genus, the Filistatidae. There is a bit more information that can possibly be obtained from the existing data, such as deriving inferences from size differences of burrows in polluted vs control areas. In justifying their approach towards this study, the authors are encouraged to refer to the broader literature on investigations of terrestrial pollution, e.g. Wiens, J. A., and K. R. Parker. 1995. Analyzing the effects of accidental environmental impacts: approaches and assumptions. Ecological Applications 5:1069-1083 (and more recent publications).
Title: By long-term, decadal time scales are implied. The monitoring was not so much long-term as the impacts. Also, population recovery was not investigated. The title could therefore be rephrased to read, e.g.: Five-year monitoring of a desert-dwelling spider following an environmental disaster indicates long-term impacts.
Line 589: explain that there is a possibility that they could naturally be less abundant in the low-lying ground where oil flowed, as these would previously have been more disturbed by water flooding. This could only have been tested with a before and after study, but it was not possible to predict the accidental oil spill.
Line 615: If burrow diameter relates to spider size (this can perhaps be determined from the spiders collected in the field where both parameters were measured), then smaller burrow size in oiled areas would indicate that the spiders were smaller, which could be the result of less food being available to these spiders in the oil-polluted areas. The above-mentioned relationship should be tested.
Line 625: you could add that it could perhaps have depressed the density of prey available to these sit-and-wait predators, e.g. if ants avoid going over oil-contaminated soil, the spiders encounter fewer ants as prey, thus depressing reproduction. Whitford considered ants to be keystone species in deserts, e.g. Whitford, W. G. 2000. Keystone arthropods as webmasters in desert ecosystems. Pages 25-41 Coleman, D.C.
Hendrix, P.F. CABI Publishing, Oxon, U.K.
Alternatively, there are many papers on that subject by that author (and others), although I am not sure whether any concern oil-polluted ecosystems.
More detailed comments are inserted into the attached PDF copy of the manuscript.

Author Response
Response: We are grateful to the academic editor and the two reviewers for the positive and important reviews of our manuscript. An acknowledgement of these efforts has been added to the manuscript. In this re submission, we have addressed all comments. Our specific responses are herein.
Reviewer 1
This study is an important contribution towards understanding the impacts of oil pollution, and also adds rarely recorded information on an unusual spider family/genus, the Filistatidae. There is a bit more information that can possibly be obtained from the existing data, such as deriving inferences from size differences of burrows in polluted vs control areas. In justifying their approach towards this study, the authors are encouraged to refer to the broader literature on investigations of terrestrial pollution, e.g. Wiens, J. A., and K. R. Parker. 1995. Analyzing the effects of accidental environmental impacts: approaches and assumptions. Ecological Applications 5:1069-1083 (and more recent publications).
Response: we appreciate the suggestions of the reviewer and for pointing out the paper by Wiens & Parker 1995; we refer to the specific comments and have added the suggested reference.
Title: By long-term, decadal time scales are implied. The monitoring was not so much long-term as the impacts. Also, population recovery was not investigated. The title could therefore be rephrased to read, e.g.: Five-year monitoring of a desert-dwelling spider following an environmental disaster indicates long-term impacts.
Response: We accepted the suggestion and changed the title accordingly.
Detailed comments are inserted into the attached PDF copy of the manuscript.
Page 1 line 40: I suggest avoiding talking about recovery, as such a trend was not investigated. Rather speak of long-lasting impacts.
Response: We accepted the suggestion and changed accordingly.
Page 2 line 52: Later this is described as five years but is actually more like 4.5. If they are summer-active, then it would be 5 summers. My point is to report this consistently in this paper.
Response: We accepted the suggestion and changed accordingly to 5.
Page 3 line 107: species?
Response: we added “several species of the genus” to make it clear we talk about the genus and not a specific species.
Page 3 line 114: Delete. This statement is repeated below, and is better positioned below
Response: we deleted as suggested.
Page 4 line 154: these are shown as blue dots, not black
Response: corrected accordingly to blue.
Page 4 line 171: these are shown as blue dots, not black
Response: corrected accordingly to blue.
Page 4 line 171: this looks very high for an average (24/7/365) - maybe the the average maximum is meant?
Response: We rechecked the paper we cited (Goldreich and Karni 2001) and the citation is ok, but as this average is indeed high – we checked several other sources and added a citation and corrected it to 50 Celsius, according to Amit et al 1999.
Amit, R.; Zilberman, E.; Porat, N.; Enzel, Y. Relief Inversion in the Avrona Playa as Evidence of Large-Magnitude Historical Earthquakes, Southern Arava Valley, Dead Sea Rift. Quat. res. 1999, 52, 76–91, doi:10.1006/qres.1999.2050.
Page 5 line 207: the detailed description below accounts for only 24 quadrats
Response: we corrected the text.
Page 5 line 221 – figure 2: consider rearranging so that this is A, B, C (moving D up to position B, E up to position C) so that pictures from the two sites are grouped
Response: changed as suggested.
Page 5 line 223-4: change females to female spiders
Response: added “spiders”.
Page 6 line 227: change dig to excavated
Response: changed to “excavated”.
Page 6 line 235: remove s from “hours”
Response: removed “s”.
Page 6 line 248: delete “were”
Response: removed “were”.
Page 6 line 253: delete “,”
Response: deleted “,”.
Page 8 line 271: briefly state what these limits are, whether geographic distribution, or based on morphological parameters, or something else
Response: we rephrased this to clarify that we mean taxonomic limits.
Page 10 line 314-5: rephrase, I do not understand this sentence
Response: We rephrased the sentence.
Page 11 line 346: add statement that the arrow in D points to the spider location in the experiment
Response: added.
Page 12 line 363: these potentially occupied burrows should perhaps be separated into a third category. Well-fed or moulting spiders may not respond to vibrations at their entrance, but fresh silk could suggest presence.
Response: we re-assigned the burrows into three categories: spider present; active burrow, but spider not seen; no spider or unknown – and changed the graph and test accordingly.
Page 12 line 368: change along to “during”.
Response: changed to “in”.
Page 13 line 408: insert: "24-h experiment".
Response: added.
Page 14 line 410: remove s from “hours”
Response: removed “S”.
Page 14 line 434: I have never heard of gibsum. Perhaps you mean gypsum
Response: corrected.
Page 16 line 496: change is to “was”
Response: changed.
Page 16 line 497: Refer to Magalhael having noted that some other species from this genus appear to have skewed sex-ratios (and also did not find adult males for some species). In the discussion, you could suggest some reasons why no adult males were found in the current study, e.g., it could be because males no longer inhabit burrows as adults when they seek mating opportunities with females, or perhaps because males mature in a restricted breeding season (give references from other species)
Response: we added several sentences on page 19 under “3.4.2. Natural history, including citation of Magalhaes et al 2020:
“No males were documented, but in many spider families, males would be caught only in pitfall traps (during mate search). As part of the monitoring, we used pitfall traps, but only for two weeks (in early summer and late summer), and we did not collect any male Sahastata. In addition, it was suggested for other Sahastata species that males are short lived after maturity, and in captivity they account for 20-25% of the juveniles raised.”
Page 18 line 531: do you mean "variety"?
Response: corrected.
Page 18 line 531: " covered"
Response: corrected.
Page 18 line 532: "gypsum”
Response: corrected.
Page 18 line 532: "trees”
Response: corrected.
Page 18 line 534: change word order: to probably spun
Response: changed to: Nests were probably constructed …
Page 18 line 534: correct to vegetation
Response: corrected.
Page 18 line 536: change word order: were 7-15 cm in length
Response: corrected.
Page 18 line 538: correct to horizontal
Response: corrected.
Page 18 line 539: correct to parallel
Response: corrected.
Page 18 line 539: correct to gypsum
Response: corrected.
Page 18 line 539: correct to separating
Response: corrected.
Page 18 line 539: delete between
Response: deleted.
Page 18 line 540: change to from
Response: changed.
Page 18 line 541: change to a few
Response: changed.
Page 18 line 542: captivity
Response: changed.
Page 18 line 542: for two to three
Response: changed.
Page 18 line 542: mainly
Response: corrected.
Page 18 line 546: insert comma
Response: added.
Page 18 line 546: mainly
Response: corrected.
Page 18 line 547: insert comma
Response: added.
Page 18 line 549: insert comma
Response: added.
Page 18 line 550: insert comma
Response: added.
Page 19 line 551: assemblage
Response: corrected.
Page 19 line 551: sessile
Response: corrected.
Page 19 line 551: percentage or proportion
Response: changed to proportion.
Page 19 line 551: replace "nests are" with "prey remains is"
Response: replaced.
Page 19 line 564: delete dash
Response: deleted.
Page 20 line 575: I suggest referring to another study at the same locations: Ferrante, M., D. Möller, G. Möller, E. Menares, Y. Lubin, and M. Segoli. 2021. Invertebrate and vertebrate predation rates in a hyperarid ecosystem following an oil spill. Ecology and Evolution 2021:1-8.
Response: Ferrante et al 2021 investigated predation by mobile predators, not sedentary, burrow-dwellers such as Sahastata, and therefore this paper is not relevant to our findings. In addition, we deleted the first sentence.
Page 20 line 580: replace ", along four-year monitoring," with "during four years of monitoring"
Response: replaced.
Page 21 line 585: add a caveat to the effect that this was an after-the-event investigation that could not follow standard experimental procedures (you could refer to Wiens, J. A., and K. R. Parker. 1995. Analyzing the effects of accidental environmental impacts: approaches and assumptions. Ecological Applications 5:1069-1083."
Response: We revised accordingly and added the suggested reference.
Page 21 line 589: explain that there is a possibility that they could naturally be less abundant in low-lying ground where oil flowed, as these would previously have been more disturbed by water flooding. This could only have been tested with a before and after study, but it was not possible to predict the accidental oil spill.
Response: we added these sentences: “It is also possible that S. aravensis is less abundant in the contaminated areas due to disturbance by naturally occurring floods rather than the oil flow. Indeed, we found burrows that apparently had been destroyed during seasonal flooding; however, these occurred in both control and oil-contaminated plots. Furthermore, the wadis in both areas are shallow and flood waters do not necessarily follow the trajectory of the oil spill. Thus, flooding is unlikely to solely explain the significant difference in burrow numbers in contaminated and clean soils”
Page 21 line 603: how deep did the oil penetrate? Could the spider burrows extend below the oil-soaked layer?
Response: The oil penetrated maximum 30 cm. So, the answer is - no. – we added this information in the introduction: “It was estimated that the oil penetrated between few millimeters to a maximum of 30 cm in both 1975 and 2014 sites”
Page 21 line 613: Were there burrowing isopods such as Hemilepistus reaumuri at the study sites? If so, they could play a facilitating role for these spiders. Either way, the presence or absence of these burrowers could be mentioned, given that they are well-publicized.
Response: There are no burrowing Hemilepistus reaumuri or any other burrowing isopods in the Arava. We added: “..nests or other empty burrows such as those of small lizards or empty ant nests.”
Page 21 line 615: If burrow diameter relates to spider size (this can perhaps be determined from the spiders collected in the field where both parameters were measured), then smaller burrow size in oiled areas would indicate that the spiders were smaller, which could be the result of less food being available to these spiders in the oil-polluted areas. The above-mentioned relationship should be tested.
Response: we tested the correlation between spider size and burrow opening size and found no significant correlation. We added this in 3 places:
- in the results: “3.3. Soil preference laboratory experiments. The cephalothorax length of the ten spiders used for the experiments ranged between 4.2 mm to 5.5 mm, and their maximum original burrow opening length ranged between 1.2 cm to 3 cm, with no significant correlation between the cephalothorax and original burrow sizes (correlation coefficient = -0.59, p=0.07; Appendix A2). In all trials, spiders were active during night, and did not move much during daytime. “
- In Appendix 2
- In the discussion: “It may be suggested that Sahastata burrows were smaller due to smaller size of spiders, which is a result of lower availability of prey. However, we found no significant correlation between spider size and burrow opening size for the ten spiders collected for the laboratory experiment. In addition, some ants, that are probably the main prey of S. aravaensis, were significantly more abundant in the 1975 oil-contaminated plots [46]. Similar pattern of ants high abundance in oil-contaminated soils was also found in Saltmarshes in the USA [47], and in a desert in Kuwait [48].
Page 21 line 622: how thick was the layer?
Response: The oil penetrated maximum 30 cm. We added this information in the introduction: “It was estimated that the oil penetrated between few millimetres to a maximum of 30 cm in both 1975 and 2014 sites”
Page 21 line 625: you could add that it could perhaps have depressed the density of prey available to these sit-and-wait predators, e.g. if ants avoid going over oil-contaminated soil, the spiders encounter fewer ants as prey, thus depressing reproduction. Whitford considered ants to be keystone species in deserts, e.g. Whitford, W. G. 2000. Keystone arthropods as webmasters in desert ecosystems. Pages 25-41 Coleman, D.C.
Hendrix, P.F. CABI Publishing, Oxon, U.K.
Alternatively, there are many papers on that subject by that author (and others), although I am not sure whether any concern oil-polluted ecosystems.
Response: we rewrote the whole paragraph: “Burrow opening diameter of S. aravaensis was significantly smaller in 1975 oil-contaminated soil quadrats compared to other quadrats (Figure 7C). We suggest that spiders in the oil-contaminated quadrats were unable to expand their burrows due to the nature of the soil in the untreated spill areas. Although we did not find greater soil hardness in the 1975 oil-contaminated soil survey quadrats, the measurements were only of the soil crust, and not of deeper parts of the soil, where digging may be more difficult due to a layer of solidified oil. Other changes in soil characteristics, such as extreme hydrophobicity [10] that resulted from the oil spill might affect the spiders’ ability to expand their burrows. Additional explanations could be a lower availability of larger empty burrows or smaller spider size in the oil-contaminated areas due to lower prey availability. However, burrow diameter and spider size were not correlated (see Results), and the likely main prey of S. aravaensis – ants - were significantly more abundant in the 1975 oil-contaminated plots than the control plots [46]. A similar pattern of high ant abundance in oil-contaminated soils was found also in saltmarshes in the USA [47], and in a desert in Kuwait [48].”
Page 22 line 664: assisting with
Response: corrected.
Page 22 line 665: assisting
Response: corrected.
Page 22 line 667: assistance
Response: corrected.

Reviewer 2 Report
I have with great interest read the manuscript entitled ‘Long-term monitoring of a desert burrow-dwelling spider following an environmental disaster suggests slow recovery’, which looks at spider abundance and behaviour following historic oil spills in the ‘Avrava valley in Israel. Not only do the authors look at spider abundances in the field over a 5-year period, but they supplement this with behavioural preference studies in the lab and a taxonomic species description of this new species. The study is very well designed with a comprehensive and well written manuscript. It makes a very welcome contribution to our limited understanding of the effects of anthropogenic disasters in extreme terrestrial habitats. I only have some relatively minor comments that I have added directly to the manuscript in the attached document.

Author Response
We are grateful to the academic editor and the two reviewers for the positive and important reviews of our manuscript. An acknowledgement of these efforts has been added to the manuscript. In this resubmission, we have addressed all comments. Our specific responses are herein.
Reviewer 2
I have with great interest read the manuscript entitled ‘Long-term monitoring of a desert burrow-dwelling spider following an environmental disaster suggests slow recovery’, which looks at spider abundance and behavior following historic oil spills in the ‘Avrava valley in Israel. Not only do the authors look at spider abundances in the field over a 5-year period, but they supplement this with behavioral preference studies in the lab and a taxonomic species description of this new species. The study is very well designed with a comprehensive and well written manuscript. It makes a very welcome contribution to our limited understanding of the effects of anthropogenic disasters in extreme terrestrial habitats. I only have some relatively minor comments that I have added directly to the manuscript in the attached document.
Page 3: Left hand map: The light blue squares on this figure are very hard to see, and the text is too small. A scale bar is missing.
Right hand side. Scale bar and lat long values are too small to be readable.
Response: we improved the color, enlarge the text size and scale bar, and added a scale bar.
Page 4 line 192: Vicinity of...is a bit vague. Please give mean distances.
Response: We rewrote the sentence to clarify the what we meant by vicinity: “Control plots were situated tens to maximum of 250 meters in parallel to the oil-contaminated plots…”
Page 4 line 194-5: When exactly in the year were these carried out?
Response: we added in the appendix a table with all monitoring events.
Page 6 line 231: What was the relative humidity in the room (and the one below)?
Response: we did not measure the relative humidity in the room at the time of the experiment.
Page 6 line 248: Delete “were”
Response: removed “were”.
Page 7 figure 3: Nice and very relevant photos, but a scale bar would be nice (if you measured burrow size of these, you could perhaps use that to construct a scale bar?)
Response: Unfortunately, those burrows were not taken with a scale, creating a scale from burrow measurements will not be accurate because of the different angle.
Page 8 figure 4: Again, nice photos, but less relevant, so consider removing or move to the appendix.
Response: we think the photos of how spider presence was evaluated are important and we decided to leave it in the main text.
Page 9 figure 5: Again nice photos, but less relevant, so consider removing or move to the appendix.
Response: we think the photos of the quadrat are important to show how this was done in the field and we decided to leave it in the main text.
Page 10 2.5.3.: I don't fully follow this explanation. Isn't final location the response variable? What are the fixed effect(s)?
Response: we corrected and rephrased the sentence: “We used logistic regression (with logit link function) to test the effect of the initial location of the spider as the main factor, and the spider identity and trial number as random factors, on the final location of the spider in the arena.”
Page 12 figure 7: In order to cut down on the number of figures further (see also other comments), you could consider combining figures 7, 8 and 9 into one 1 x 3 figure.
Response: we accepted this suggestion.
Page 13 figure 9: units missing on the y-axis.
Response: we added the units.
Page 13 figure 10: The resolution of the text on this figure is very poor and I cannot read it.
Response: We improved the resolution.
Page 18 line 542: correct spelling for captivity and mainly
Response: corrected.
Page 20 figure 15: Figure not needed.
Response: we think the photos of the burrow excavation are important and we decided to leave it in the main text.
Page 21 lines 603-614: This section is very interesting, but I'd like a bit more details. So which organisms would create these brurrows? I'm not familiar with this famly, but could the males (which I gather are unknown at this stage) potentially have some adaptations for digging and female use burrow quality to aid their choice of males as in fiddler crabs?
Response: We do not know who makes the original burrows. Not a lot is known about the genus – however, no morphological adaptation was found for digging in males or females, we clarified this point.
